# Barriers and facilitators of successful deprescribing as described by older patients living with frailty, their informal carers and clinicians: a qualitative interview study

George Peat ,[1,2,3] Beth Fylan ,[2,3,4] Iuri Marques,[3] David K Raynor ,[5] Liz Breen ,[2,3] Janice Olaniyan,[1,2,3] David Phillip Alldred [2,5]

¹School of Health and Life Sciences, University of Bradford, Bradford, UK
²Yorkshire & Humber Patient Safety Translational Research Centre, NIHR, Bradford, England
³School of Pharmacy and Medical Sciences, University of Bradford, Bradford, UK
⁴Bradford Institute for Health Research, Yorkshire Quality and Safety Research Group, Bradford, UK
⁵School of Healthcare, University of Leeds, Leeds, UK

**Correspondence to**
Dr George Peat;
gwpeat@bradford.ac.uk

## ABSTRACT

**Objective** To explore the barriers/facilitators to deprescribing in primary care in England from the perspectives of clinicians, patients living with frailty who reside at home, and their informal carers, drawing on the Theoretical Domains Framework to identify behavioural components associated with barriers/facilitators of the process.

**Design** Exploratory qualitative study.

**Setting** General practice (primary care) in England.

**Participants** 9 patients aged 65+ living with frailty who attended a consultation to reduce or stop a medicine/s. 3 informal carers of patients living with frailty. 14 primary care clinicians including general practitioners, practice pharmacists and advanced nurse practitioners.

**Methods** Qualitative semistructured interviews took place with patients living with frailty, their informal carers and clinicians. Patients (n=9) and informal carers (n=3) were interviewed two times: immediately after deprescribing and 5/6 weeks later. Clinicians (n=14) were interviewed once. In total, 38 interviews were undertaken. Framework analysis was applied to manage and analyse the data.

**Results** 6 themes associated with facilitators and barriers to deprescribing were generated, respectively, with each supported by between two and three subthemes. Identified facilitators of deprescribing with patients living with frailty included shared decision-making, gradual introduction of the topic, clear communication of the topic to the patient and multidisciplinary working. Identified barriers of deprescribing included consultation constraints, patients' fear of negative consequences and inaccessible terminology and information.

**Conclusions** This paper offers timely insight into the barriers and facilitators to deprescribing for patients living with frailty within the context of primary care in England. As deprescribing continues to grow in national and international significance, it is important that future deprescribing interventions acknowledge the current barriers and facilitators and their associated behavioural components experienced by clinicians, patients living with frailty and their informal carers to improve the safety and effectiveness of the process.

## Strengths and limitations of this study

► A strength and novel aspect of the study was interviewing patients about their deprescribing journey at two time points, immediately after their deprescribing consultation and 5 to 6 weeks later.

► A strength of the study was the inclusion of patients, their informal carers and healthcare professionals collectively, providing a multiperspective insight into the barriers and facilitators of deprescribing in primary care in England.

► Our sample of patients and informal carers were lower than expected, due to challenges recruiting patients who were actively having a medicine deprescribed.

## INTRODUCTION

Frailty can be understood as the age-related decline in physiological structures that can result in, among other consequences, an increased vulnerability to adverse drug reactions (ADRs). This is due to the interplay between physiological changes and pharmacokinetics and pharmacodynamics.[1] Consequently, there is an increasing focus on the safe and appropriate use of medicines in patients living with frailty. One strategy to improve medicines safety is to reduce or stop medicines that may no longer be appropriate or beneficial, a process referred to as deprescribing.[2]

If undertaken appropriately, deprescribing has been shown to support a reduction in ADRs and can improve quality of life.[3] Several process tools to support deprescribing have been created.[2 4–6] However, to date, no implementation science frameworks exist specific to deprescribing. Such a framework may allow researchers to implement deprescribing, with the knowledge of the

contextual determinants in a specific setting, to cater the process to the specific needs of the practice population.[7] Furthermore, identifying factors that facilitate or hinder deprescribing continue to be a high priority to ensure the process of deprescribing is safely and effectively implemented to optimise patient outcomes.[8]

Evidence suggests that barriers to deprescribing exist at various socioecological levels.[8] For instance, several studies have found clinicians in primary care face time restraints that hinder effective deprescribing, including limited time to review medicines, engage in consultations and provide adequate follow-up.[9–11] Interpersonal barriers also exist and include ineffective communication with prescribers across healthcare settings, with some prescribers hesitant to deprescribe medicines prescribed by specialists in other care settings.[10 12 13] Challenges with communicating deprescribing effectively to patients using appropriate discourse exist.[12] Similarly, studies have reported prescriber hesitancy to deprescribe due to limited knowledge and guidance on how to do so appropriately and safely.[9 14 15] Finally, clinician hesitancy to deprescribe also exists because of a fear that deprescribing may be perceived by the patient as a withdrawal of care.[9]

From the perspective of patients living with frailty, managing multiple morbidities often means engaging with multiple prescribers over a sustained period, with evidence suggesting this scenario can make deprescribing challenging. For instance, an increased risk of poor communication between parties involved in a patient's care has been cited, resulting in patients being confused about which prescriber has authority to deprescribe.[10] Similarly, patients prescribed medicines for a significant period are often reluctant to stop those medicines, particularly if they believe that the medicine is necessary, or that stopping the medicine would result in negative outcomes.[9 16] Likewise, older adults often lack sufficient knowledge about their medicines, limiting confidence and willingness to engage in decision-making related to deprescribing.[15] Finally, paternalistic prescribing attitudes, void of shared decision-making, have also been shown to be a barrier to deprescribing.[15 16]

Studies have also reported facilitators of deprescribing. These include ensuring clinicians receive sufficient information and support related to the risks and benefits of deprescribing specific medicines to enable an informed decision.[14] Similarly, adopting an organisational culture, which encourages peer-to-peer learning, has been found to improve the self-efficacy of clinicians to deprescribe.[12 17] From a patient perspective, factors such as shared decision-making, trust in the clinician as a result of familiarity built through sustained clinician–patient relations and a desire to reduce medicines have been found to support deprescribing.[10 18–20]

Despite a growing evidence base on barriers to and facilitators of deprescribing, studies largely exist outside of the UK with only 3 out of 40 studies identified in a recent systematic review on deprescribing in primary care.[8] More so, there is a paucity in research into deprescribing focused specifically on patients living with frailty, a patient cohort increasingly considered a priority group for interventions that improve medicines safety. For example, recent policy in England, such as the General Medical Services contract, requires general practice (GP) to conduct structured medication reviews (SMR) with patients living with moderate to severe frailty.[21] A potential outcome of an SMR is that a patient may have one or more of their medicines stopped. To support the success and effectiveness of deprescribing as a potential outcome of SMRs for patients living with frailty, clinicians need to be aware of the barriers to and facilitators of deprescribing in this group. Furthermore, while international studies have aimed to understand the patient perspective on deprescribing, no studies appear to have captured the barriers/facilitators of deprescribing from the perspective of patients who were actively having a medicine reduced or stopped. Finally, few studies have identified the determinants of behaviour that either facilitate or act as a barrier to deprescribing. To address these gaps this study aimed to explore the barriers/facilitators to deprescribing from the perspectives of clinicians, patients living with frailty who reside at home and were candidates for deprescribing and their informal carers, in the specific context of primary care in England, drawing on the Theoretical Domains Framework (TDF) to identify behavioural components associated with barriers/facilitators of the process.

## METHODS
### Study design, setting and sampling (inclusion/exclusion)
The study adopted a qualitative design and recruited from four GP practices across regions of Yorkshire and Humber in England. Purposeful sampling was used to recruit patients ≥65 years old, living with frailty (either formally diagnosed and registered in the primary care electronic health record or at risk of frailty as defined by the electronic Frailty Index, embedded within GP prescribing systems)[22] who attended a consultation to deprescribe. In addition, informal carers who provided informal care to patient participants were also recruited to the study. Clinicians recruited eligible patients if, during their appointment, a medicine had been identified for deprescribing. Patients were provided an invitation letter and participant information sheet. Those who agreed to participate gave signed, informed consent. Patients were asked if they had an informal carer and invitations were extended to them.

To capture the perspective of clinicians, healthcare professionals involved in deprescribing (general practitioners [GPs], pharmacists, nurses and pharmacy technicians) were interviewed. Clinicians were recruited from the four participating surgeries. They were approached by a practice representative and were invited to take part, received an invitation letter and participant information sheet. Those who agreed to participate gave signed, informed consent. All clinicians had experience of deprescribing.

This qualitative study is part of a larger programme of work aimed at enhancing the process of deprescribing with patients living with frailty in the primary care setting in England, undertaken by the 'Safe Use of Medicines' research theme, part of the National Institute for Health Research

(NIHR) Yorkshire & Humber Patient Safety Translational Research Centre.

## Data collection

Semistructured interviews were conducted between September 2018 and February 2019. Participants provided consent for the interviews to be audio-recorded and transcribed verbatim. Interviews with clinicians guided by an interview schedule (online supplemental file 1 'interview schedule clinicians') took place at their practice and explored their experiences of deprescribing. Four interview schedules (online supplemental files 2-5 'interview schedule patients first int', 'interview schedule patients second int', 'interview schedule informal carers first int'. 'interview schedule informal carers second int'), informed by the literature and the TDF, and developed in consultation with a patient public involvement and engagement (PPIE) member, were designed to capture experiences of deprescribing and potential barriers and facilitators during the process from the perspective of patients and their informal carers.[23] The TDF, which is a theoretical framework used to understand behavioural influences, was used to focus and facilitate discussions about the determinants of behaviours of clinicians, patients and their informal carers. Patients and their informal carers were interviewed two times at their homes, up to 1 week after their appointment, and 5–6 weeks later. The first interview explored patients' immediate perceptions of deprescribing, while the second focused on outcomes and their experience of the process. Interviews lasted on average 30 min. Interviews were conducted by experienced healthcare researchers from a range of backgrounds: a psychologist experienced in pharmacy practice and patient safety research, a sociologist and a pharmacist. All interviewers used the same interview guide and regular discussions were held to compare experiences of the interviews and discuss collected data.

## Data analysis

Framework analysis was applied to analyse the data.[24] The approach provides a practical and structured approach to data management and analysis through five stages: data familiarisation; iterative development of a framework for the purpose of data management; indexing of individual transcript data to the categories within the framework; summarising the data within each category and interpretation of the data.[24] Data were coded by GP, BF and IM, with LB, DPA, DKR and JO providing feedback on the coding framework. Two broad master themes representative of the barriers and facilitators to deprescribing experienced by participants were first identified. Each broad theme was then explored to identify themes and subthemes representative of a barrier or facilitator of deprescribing. Finally, each subtheme was mapped to the TDF to identify domains to target for deprescribing interventions. Data were managed using Microsoft Excel.

## Patient and public involvement

All patient-facing recruitment materials, including the topic guide/s, were reviewed by a PPIE representative. The PPIE representative was an experienced local lay contributor and a lay leader within the NIHR Yorkshire and Humber Patient Safety Translational Research Centre.

## RESULTS

Thirty-eight interviews in total were undertaken. Nine patients were interviewed at two separate points, up to 1 week after appointment and 5 to 6 weeks later, along with three informal carers (table 1). Fourteen clinicians were also interviewed once, comprising of six GPs, two practice pharmacists, five practice nurses and advance nurse practitioners and one pharmacy technician (table 2). Analysis of the data identified three themes associated with facilitators of deprescribing, with each theme supported by between two and three subthemes (figure 1). Three themes associated with barriers of deprescribing were also identified and were similarly supported by between two and three subthemes (figure 2). Tables 3 and 4 illustrate the themes and subthemes identified, with each mapped onto the TDF to identify behavioural components associated with barriers and facilitators of deprescribing. Themes and subthemes associated with facilitators to deprescribing were attributed to the following TDF domains: *Skills, Intentions, Belief about capabilities, Knowledge, Social influences, Social/professional role and*

**Table 1** Demographics: patients and informal carers

| Number | Patient code | Age | Gender | Ethnicity | Number of medicines before deprescribing | Informal carers | |
| --- | --- | --- | --- | --- | --- | --- | --- |
| | | | | | | Code | Relationship |
| 1 | Practice 1-Patient 1 | Early 70s | Male | White British | 5 | N/a | |
| 2 | Practice 1-Patient 2 | Late 80s | Male | White British | 15 | N/a | |
| 3 | Practice 1-Patient 3 | Late 80s | Male | White British | 8 | N/a | |
| 4 | Practice 2-Patient 1 | Early 90s | Male | White British | 7 | N/a | |
| 5 | Practice 2-Patient 2 | Mid 80s | Male | White British | 5 | N/a | |
| 6 | Practice 3- Patient 2 | Early 70 s | Female | White British | 15 | N/a | |
| 7 | Practice 3-Patient 3 | Late 70 s | Female | White British | 9 | Practice 3 -IC3 | Daughter |
| 8 | Practice 4-Patient 1 | Late 70 s | Female | White British | 16 | Practice 4-IC1 | Daughter-in-law |
| 9 | Practice 4-Patient 3 | Late 70 s | Male | White British | 14 | Practice 4-IC3 | Wife |

GP, general practitioner.

| Table 2 | Demographics: clinicians | | | | | |
|---|---|---|---|---|---|---|
| Number | HCP code | Gender | Ethnicity | Job role | Prescriber | Years and type of experience |
| 1 | Practice 1-HCP1 | Female | White British | GP | Yes | 15 years as a doctor, 8 years as a GP |
| 2 | Practice 1-HCP2 | Female | White British | Practice pharmacist | No | 34 years as a pharmacist, 9 months in primary care |
| 3 | Practice 1-HCP3 | Female | White British | Practice nurse | Yes | 23 years as a nurse, 14 years in primary care, 15 months as prescriber |
| 4 | Practice 2-HCP1 | Female | White British | GP | Yes | 25 years as a doctor, 22 years as a GP |
| 5 | Practice 2-HCP2 | Female | White British | Pharmacy technician | No | 32 years as pharmacy technician, 6 years in primary care |
| 6 | Practice 2-HCP3 | Female | White British | Advanced nurse practitioner | Yes | 20 years as a nurse, 15 years in primary care, 3 years as prescriber |
| 7 | Practice 3-HCP1 | Female | White British | Advanced nurse practitioner | No | 44 years as a nurse, 10 years in primary care |
| 8 | Practice 3-HCP2 | Female | White British | GP | Yes | 28 years as a doctor, 23 years as a GP |
| 9 | Practice 3-HCP3 | Male | White British | GP | Yes | 11 years as a doctor, 6 years as GP |
| 10 | Practice 3-HCP4 | Female | White British | GP | Yes | 20 years as a doctor, 14 years as GP |
| 11 | Practice 4-HCP1 | Female | White British | GP | Yes | 7 years as a doctor, 1.5 years as a GP |
| 12 | Practice 4-HCP2 | Female | White British | Practice nurse | Yes | 29 years as a nurse, 12 years as practice nurse, 8 years as a prescriber |
| 13 | Practice 4-HCP3 | Female | White British | Practice nurse | Yes | 4 years as a nurse, 2 years in primary care and as prescriber |
| 14 | Practice 4-HCP4 | Female | White British | Practice pharmacist | No | 13 years as a pharmacist, 2.5 years in primary care |

GP, general practitioner.

*identity, Environmental context and resource, Memory, attention and decision processes, Goals.* Conversely, the themes and subthemes associated with barriers to deprescribing were attributed to the following domains: *Social influences, Environmental context and resources, Social/professional role and identity, Memory, attention, and decision processes, Skills (interpersonal), Emotion.*

Each whole theme and their subthemes representative of a barrier, or facilitator of successful deprescribing, are summarised in the next section.

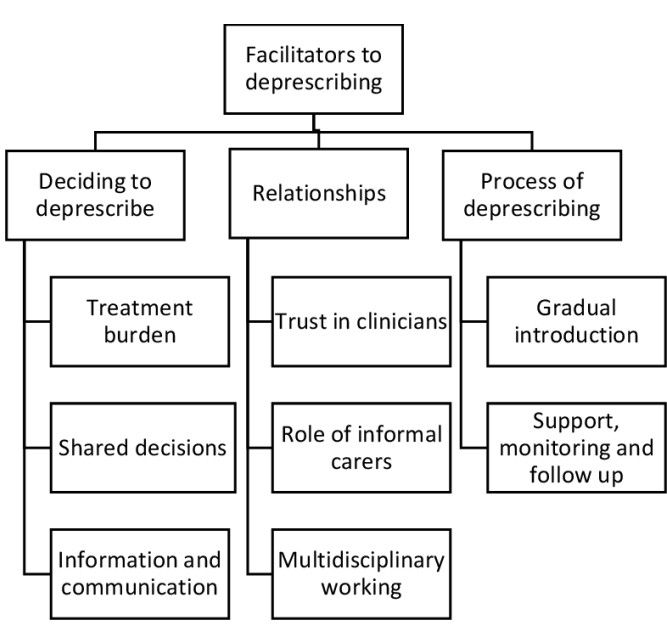

**Figure 1** Diagram mapping the themes and subthemes developed related to facilitators of deprescribing.

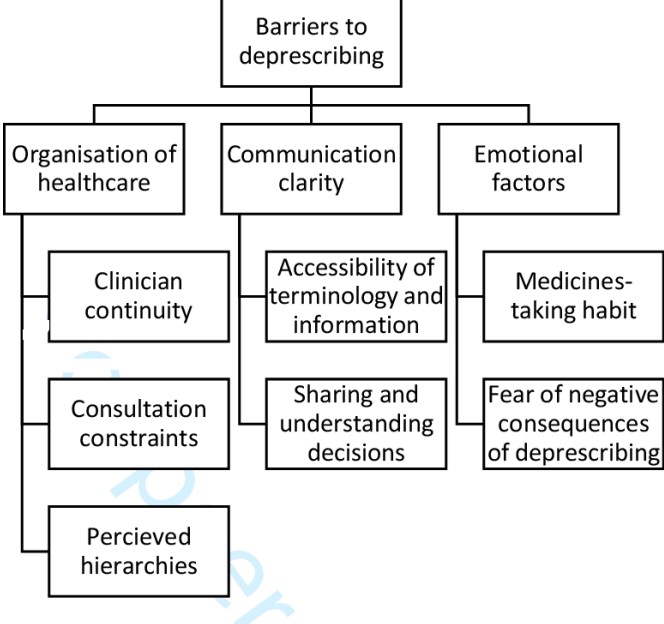

**Figure 2** Diagram mapping the themes and subthemes developed related to barriers of deprescribing.

**Table 3** Facilitators of deprescribing

| Theme | Subtheme | TDF domain | Sample quotes |
|---|---|---|---|
| Deciding to deprescribe | Treatment burden | Intentions | "I'm glad those things have reduced." P2 (practice 1)<br>"We get quite a few people who are on a lot of medication, don't generally know why they take certain ones." HCP4 (practice 4)<br>"And you do get a lot of them saying is there any of these that I could actually stop?" HCP2 (practice 4). |
| | Shared decisions | Belief about capabilities<br>Skills (ability) | "I am a great believer in shared decision making so I do spend a lot of my consultations around making sure that [the patient knows the reason for taking the medicine], because their concordance [adherence] will be very poor if they don't understand why they are taking it and they need to understand the importance." HCP3 (practice 1). |
| | Information and communication | Skills (interpersonal)<br>Knowledge (procedural) | "…you're doing something that they ultimately disagree with and you're telling them that they can't have something that they want, that's the crux of the matter, but you also have to use your communication skills to explain why you're doing that, why is it that you're stopping this medication." HCP3 (practice 3). |
| Relationships | Trust in clinicians | Skills (interpersonal) | "So, I think for this lady I knew her, so I had a relationship with her which helps. And I think building that understanding and trust between us, so I think it was positive that we had trust…" HCP4 (practice 4) |
| | Role of informal carers | Social Influences (social support) | "She can't always take in what's being said. If I had known that they were going to have her in to take her medication away I'd have gone with her, so that I could listen to the explanations that were being given to her to allay her worries." IC1 (practice 4) |
| Relationships (cont) | Multidisciplinary working | Social/professional role and identity (organisational commitment)<br>Environmental context and resource (organisational culture) | "We've done a lot of deprescribing work here and it has been successful because we decided to involve the whole team. It would be no good if I sat here in a practice with six GPs and decided to improve my deprescribing because the other five GPs would be quietly sabotaging that without realising because we're not working together. If we all change our deprescribing patterns, then we have a hope of really making (a) change." HCP1 (practice 1) |
| Process of deprescribing | Gradual introduction | Skills (interpersonal)<br>Memory, attention and decision processes<br>Reinforcement<br>Beliefs about consequences<br>Goals<br>Skills (competence) | "It was about a control thing, they start off with one thing, one little step, and then they came back and said how they felt about that step; for this case it was pregabalin, and I talked about how to reduce it and if actually they got side effects or if things got worse it was then to speak and actually go further; just wait a little while and then make your next step, and giving them back some of the control, so they felt in charge rather than me, and that always seems to work much better." HCP2 (practice 3) |
| Process of deprescribing (cont) | Support, monitoring, follow-up | Goals (target setting)<br>Knowledge (procedural) | "On that occasion [I received] as much [support] as I'd wanted, previously I'd discussed it [discontinuing aspirin] with [my GP] and with other practitioners at the health centre and been very, felt very supported." P1 (practice 1) |

TDF, Theoretical Domains Framework.

## Facilitators to deprescribing

### Theme 1: deciding to deprescribe

Arriving at a decision to deprescribe a medicine(s) was facilitated by several factors. First, some patients wished to reduce their medicines, a viewpoint reflected both in the accounts of patients and clinicians (table 3). To reach a deprescribing decision, clinicians also advocated shared decision-making during consultation:

"I am a great believer in shared decision-making so I do spend a lot of my consultations around making sure that [the patient knows the reason for taking the medicine], because their concordance [adherence] will be very poor if they don't understand why they are taking it…" HCP3 (practice 1).

Ensuring a patient understood why deprescribing was appropriate was essential in maintaining adherence with a new regimen, with shared decision-making integral to developing this understanding. Similarly, effectively communicating why deprescribing was appropriate was important, particularly with patients who were initially hesitant to deprescribe (table 3).

**Table 4** Barriers to deprescribing

**Barriers to deprescribing**

| Theme | Subtheme | TDF domain | Sample quotes |
|---|---|---|---|
| Organisation of healthcare | Clinician continuity | Social influences Environmental context and resources | "You can't get in touch with your Doctor. It's no good going to another Doctor that doesn't understand that you're coming off your morphine and how it's affecting you." P2 (Practice 3)<br>"You never see the same doctor twice (…) unless you're under one person all the time you could go haywire (…) That's often the way when you see one doctor after another, your frequent doctor gets changed. They jump in too quick, if you're under the care of one person then that person can really monitor you." P1 (practice 2) |
| | Consultation constraints | Environmental context and resources | "We're in tough times in primary care and pressure of demand is really, really high and so ad-hoc deprescribing is really difficult to do." HCP1 (practice 1)<br>"I think it might have been easier if I hadn't done it (reviewed patient's medicines) over the phone, it might have actually been easier with that patient in front of me, but I've got time limitations." HCP2 (practice 1)<br>"They haven't got time you see; you go into the surgery and you've only 10 minutes (to see the GP)." IC3 (Practice 4) |
| | Perceived hierarchies | Social/professional role and identity | "I can't remember the exact drug, but I can tell you it was a cardiology medication, and the resistance really was they'd been told, you know this Cardiologist had told them they needed it and they needed to have it lifelong." HCP1(practice 4) |
| Communication clarity | Accessibility of terminology and information | Memory, attention and decision processes | "I think [patients] get a bit confused by the word deprescribing. It can be very much a clinical term. We need to accept that patients aren't [always familiar with clinical terms] and so I would quite often call it something different." HCP2 (practice 2)<br>"My first impression was that [deprescribing] was something that was done to one, not something that you had an active say in. It was a decision that a healthcare professional took and told the patient about it." P1 (practice 1) |
| | Sharing and understanding decisions | Memory, attention and decision processes Skills (interpersonal) | "I don't want them to say right we'll start reducing this, I need to know why and what's going to happen with me." P2 (practice 3)<br>"They don't explain what they're trying to say to you. It's as if oh we'll see you next week, go on we'll see you again later. It's…not explained what they're telling you…" P3 (practice 4). |
| Habits and fears | Medicines-taking habit | Emotion | "Because she [patient] panics. And she's been on them such long-term and it becomes habit. And I think when you stop something that you've been doing for a long time you think 'oh', you panic straight away, you think 'oh, it's all going to go wrong, it's all going to go wrong." IC1 (practice 4). |
| | Fear of negative consequences of deprescribing | Emotion | "Yes, I was afraid. Because the morphine helped me with the pain and I knew each time I was coming down I was getting pain, so I just took paracetamol and then it just kept coming down I think it was about every month she wanted me to have a go at coming down." P2 (practice 3). |

TDF, Theoretical Domains Framework.

## Theme 2: relationships

Positive relationships between involved parties were important to facilitating deprescribing. Established trust, built through a knowingness between patient and clinician, was highlighted as a facilitator by participants (table 3). In addition, involving informal carers in deprescribing consultations was cited as beneficial to assist patients where necessary to understand why their medicine/s were deprescribed. (table 3). Furthermore, the relational dynamic between clinicians and their practice was cited as significant. For instance, participants described the benefits of adopting a practice agenda to deprescribing:

"We've done a lot of deprescribing work here and it has been successful because we decided to involve the whole team. It would be no good if I sat here in a practice with six GPs and decided to improve my deprescribing because the other five GPs would be quietly sabotaging that without realising because we're

not working together. If we all change our deprescribing patterns, then we have a hope of really making [a] change." HCP1 (practice 1).

Care for older people living with frailty often involves multiple clinicians and healthcare workers. Working across teams to inform those involved in the patient's care of the decision to deprescribe was important to supporting the patient to accept their new medicine regimen:

Participant: "I've brought in the specialist, I'd already got the agreement of the GP involved in the patient's care, this was about course of action and it was highlighted to the healthcare assistant who she normally sees as well that it was, you know, the incorrect treatment for the patient." HCP3 (practice 3).

Interviewer: "What was the outcome?"

Participant: "She has now accepted that it's not the right course of treatment for her and it can actually cause more difficulties." HCP3 (practice 3).

### Theme 3: process of deprescribing

Treating deprescribing as a process, as opposed to a one-off event, was a key facilitator relayed by participants. Patients described feeling more assured and confident of the process if it was conveyed as a gradual reduction in their medicine(s). Gradually reducing dosage of medicines was associated with having a sense of control by some participants, indeed, clinicians expressed the importance of allowing the patient 'to feel in charge' of the process:

"It was about a control thing, they start off with one thing, one little step, and then they came back and said how they felt about that step; for this case it was pregabalin, and I talked about how to reduce it and if actually they got side effects or if things got worse it was then to speak and actually go further; just wait a little while and then make your next step, and giving them back some of the control, so they felt in charge rather than me, and that always seems to work much better." HCP2 (practice 3).

Deprescribing was described as 'much better' when patients were supported to feel in control, a feeling that manifested from ensuring that the patient was informed,and that steps to reduce their medicines were made tentatively in a controlled fashion:

"I think as long as you kind of go you know 'Obviously if you ever do need it again of course we'll restart it again you know, but let's just do it really slowly and we'll do it together and I'll keep an eye on you and you'll come and see me and we'll review you' I think most people are actually quite happy to at least give it a go and try and come off medication." HCP1 (practice 2).

## Barriers
### Theme 1: organisation of healthcare

Aspects of the structure and organisation of healthcare were perceived as barriers to deprescribing. Participants described difficulty in seeing the same clinician, without clinician continuity, some participants felt that they were unable to be sufficiently monitored, and their deprescribing journey would not be understood:

"You never see the same doctor twice… unless you're under one person all the time you could go haywire [frustrated]… That's often the way when you see one doctor after another. They jump in too quick. If you're under the care of one person then that person can really monitor you." Patient 1 (practice 2).

Time limitations associated with the consultation were a further barrier. Clinicians described how time demands placed on primary care did not facilitate ad hoc deprescribing. Similarly, time limitations influenced the type of deprescribing consultations clinicians offered. For example, one participant expressed that while face-to-face consultations were 'easier,' due to time limitations, reviews often took place over the telephone:

"I think it might have been easier if I hadn't done it [reviewed patient's medicines] over the phone, it might have actually been easier with that patient in front of me, but I've got time limitations." HCP2 (practice 1).

Time limitations on consultations were also voiced by informal carers, with the perception that clinicians 'haven't got the time':

"They haven't got time you see; you go into the surgery and you've only 10 minutes." IC 3 (practice 4).

The perceived hierarchy associated with clinicians was also voiced as a barrier to deprescribing. Participants spoke of this being a particular issue when patients had been prescribed medicines by a specialist in the past, who may have conveyed that the medicine(s) were a 'lifelong' regimen:

"I can't remember the exact drug, but I can tell you it was a cardiology medication, and the resistance really was they'd been told, you know this Cardiologist had told them they needed it and they needed to have it lifelong." HCP1 (practice 4).

### Theme 2: communication clarity

Challenges to communicating deprescribing effectively were identified as barriers to deprescribing. First, the term 'deprescribing' was thought to be a 'clinical term' and, therefore, created an impression that it '*was something that was done to one, not something you had an active say in*'. (P1, practice 1). Clinicians also felt that patients may feel '*a bit confused by the word deprescribing*' (HCP2, practice 2). More so, terminology was a particular barrier in settings where

the practice had *'quite a number of different populations with lots of different languages'* (HCP3, practice 2).

Communication barriers also related to the communicative make-up of the deprescribing consultation. For example, some patients felt that clinicians did not spend the time explaining why the decision to deprescribe may have been appropriate:

"They don't explain what they're trying to say to you. It's as if oh we'll see you next week. It's…not explained what they're telling you…" P3 (practice 4).

The importance of ensuring communication clarity in relation to the topic of deprescribing, particularly in the case of older people living with frailty was voiced by one clinician:

"I think sometimes people, if they're slightly older, can worry that they're just sort of, being written off."HCP2 (practice 4)

### Theme 3: habits and fears

The habitual nature of a patient's medicines regimen and fears of possible negative consequences from stopping medicines were reported to be barriers to deprescribing. For example, clinicians described how '*Some people don't like change, there's a resistance to change for some patients*' (HCP3, practice 1). Patients described feeling 'afraid' of the potential side effects of reducing or stopping medicines. Furthermore, the habitual connection to medicines also caused some patients to 'panic' when such medicines were tapered or stopped:

"Because she [patient] panics. And she's been on them such long-term and it becomes habit. And I think when you stop something that you've been doing for a long time you think 'oh', you panic straight away, you think 'oh, it's all going to go wrong, it's all going to go wrong." IC1 (practice 4).

### DISCUSSION

Several key findings were identified that contribute to the existing evidence based on deprescribing for patients living with frailty. First, and in relation to the TDF domains *environmental context and resources* and *social/professional role and identity*, adopting deprescribing as a whole practice/team agenda, whereby resource is directed towards deprescribing is likely to lead to better outcomes. More so, encouraging multidisciplinary working appears to be particularly important to successfully manage deprescribing for patients living with frailty, who often have multiple morbidities managed by multiple clinicians. Ensuring all clinicians involved in the patient's care were aware of the decision to deprescribe was found to be important in supporting the patient to accept their new medicines regimen. *Environmental and resource factors* were also found to hinder deprescribing efforts. Patients voiced frustrations related to the inability to see the same

clinician. Furthermore, while instances of successful deprescribing consultations and follow-up were reported, so too were instances of patients feeling that they did not have time to discuss the decision to deprescribe, with some patients also feeling dissatisfied about the follow-up care offered.

A key facilitator of deprescribing identified was the attribution of control to the patient throughout the deprescribing process. In relation to the TDF, domains related to *skills (interpersonal), goals* and *belief about consequences*, were identified. For example, interpersonal skills were necessary to convey an attribution of control to the patient. Likewise, where clinicians, patients and informal carers were able to agree on specific and gradual goals, patients' beliefs about the consequences of deprescribing were well managed. Furthermore, while some patients living with frailty may need to be supported by informal carers throughout the process, our findings suggest that patients benefit from feeling informed, engaged and in control of the process of deprescribing. Identifying the TDF domains associated with barriers or facilitators of deprescribing will help focus future intervention development aimed at improving deprescribing outcomes in older people living with frailty. Of the 14 TDF domains, '*environmental context and resources', 'memory, attention and decision processes'* and '*skills (interpersonal)'* were the most frequently identified by this study. Consequently, future interventions may target these behaviour domains by considering implementing prompts and cues for both healthcare professionals and patients, addressing multidisciplinary patient-centredness, and the patient's role in the deprescribing process.

Several limitations are also acknowledged. First, the number of patients and informal carers interviewed were lower than expected. Recruitment was challenging in relation to timing recruitment to when patients were having a medicine deprescribed. Similarly, the study could have benefitted from the inclusion of more informal carers. In addition, patients and their informal carers were interviewed together to support these patients to participate in the study. However, this is a potential limitation of the study as interviewing informal carers and patients together could have limited collecting the negative experiences of both due to the other being present. These limitations notwithstanding, the data provided valuable insight into the real-time barriers and facilitators experienced by clinicians, patients and their informal carers. Furthermore, a major strength and novel aspect of this qualitative study was the fact that we interviewed patients at two time points while they were *actually* having a medicine deprescribed. Interviewing patients immediately after consultation and 5-6 weeks later provided useful insight into their experience of follow-up, highlighting practices that facilitated or hindered deprescribing. However, it is also acknowledged that only interviewing patients who were actively having a medicine deprescribed, as opposed to interviewing patients living with frailty about deprescribing in general, may mean further barriers to deprescribing were not reported.

NHS England's implementation guidance for SMRs states the importance of shared decision-making and emphasises the SMR as an ongoing process as opposed to a one-off exercise.[25] Our findings support previous international evidence that highlights that patients over the age of 65 living with frailty want to be involved in decisions related to deprescribing.[19] Patients living with frailty are a priority cohort for SMRs; consequently, it is important that implementation guidance is acted on, particularly regarding ensuring patients are engaged in any deprescribing decisions made. Furthermore, while it is the case that patients may be unwilling to deprescribe if they believe their medicine is appropriate,[16] this study found some patients living with frailty were keen to engage in discussions about their medicines and how to reduce them, a finding shared elsewhere in the literature.[19] Consequently, these findings indicate that patients may be generally more open to the prospect of deprescribing than perceived.

Acknowledging the barriers to deprescribing as experienced by clinicians, patients living with frailty and their informal carers is integral to improving the effectiveness and acceptability of the process. The term 'deprescribing' was perceived by patients as either confusing, or suggestive of a process led by the clinician as opposed to being shared. Similar concerns with deprescribing discourse have also been voiced in the literature[19] suggestive of a requirement to rethink how the process is conveyed to patients and the public. To successfully deprescribe, the views and concerns of patients need to be discussed as part of the consultation.[6 16 26] Several participants voiced concerns about not understanding why their medicines were being stopped, a finding underreported in the current evidence base. Supporting patients to understand why their medicines are being deprescribed may also reduce fears and concerns about the consequences of stopping/reducing medicines, a barrier found in this study and reported elsewhere in the literature.[6 9] The shared commonalities in reported barriers/facilitators of deprescribing reported by this study and the wider literature highlight components of the process that future deprescribing intervention studies should seek to address, and those that should be prioritised to improve deprescribing effectiveness. Furthermore, identifying the barriers and facilitators of deprescribing and their associated behavioural domains can support clinicians to deprescribe safely and effectively as part of the NHS-mandated SMR programme.

In conclusion, this paper offers a timely contribution to the existing evidence base, providing insight into the barriers and facilitators to deprescribing experienced by clinicians, patients living with frailty and their informal carers within the context of primary care in England. As deprescribing continues to grow in national and international significance, it is important that future deprescribing interventions acknowledge the current barriers and facilitators and their associated behavioural components faced by clinicians, patients living with frailty, and

their informal carers to improve the safety and effectiveness of the process.

**Acknowledgements** The authors would like to acknowledge Dr Daisy Payne for her contribution in the data collection of several interviews and involvement in the very early stages of data analysis. The authors would also like to acknowledge the contribution of the wider team from the Programme Management Group of the Safe Use of Medicines Theme of the NIHR Yorkshire and Humber PSTRC.

**Contributors** BF, IM, DKR, LB, DPA all contributed to the design of the study. IM, JO were involved in data collection, and GP, IM, BF were involved in data analysis. GP drafted the manuscript. All authors contributed to the interpretation of the analysis, and critically revised and approved the manuscript. DPA is responsible for the overall content as the guarantor. GP is a senior research fellow in Patient Safety employed by the University of Bradford as part of the National Institute for Health Research (NIHR) Yorkshire and Humber Patient Safety Translational Research Centre. BF is an associate professor in Patient Safety at the University of Bradford. IM was a senior research fellow in Patient Safety employed by the University of Bradford as part of the National Institute for Health Research (NIHR) Yorkshire and Humber Patient Safety Translational Research Centre at the time of the study. DKR is emeritus professor at the University of Leeds. LB is a reader in Health Service Operations at the University of Bradford. DPA is Professor of Medicines Use and Safety at the University of Leeds. JO is a research fellow in Patient Safety employed by the University of Bradford as part of the National Institute for Health Research (NIHR) Yorkshire and Humber Patient Safety Translational Research Centre.

**Funding** This research was funded by the National Institute for Health Research (NIHR) Yorkshire and Humber Patient Safety Translational Research Centre (NIHR Yorkshire and Humber PSTRC). The views expressed in this article are those of the authors and not necessarily those of the NIHR or the Department of Health and Social Care.

**Competing interests** None declared.

**Patient consent for publication** Not applicable.

**Ethics approval** This study involves human participants and was approved by Health Research Authority (HRA) and NHS Research Ethics Committee approval was obtained (Ref. 18/YH/0140). Participants gave informed consent to participate in the study before taking part.

**Provenance and peer review** Not commissioned; externally peer reviewed.

**Data availability statement** All data relevant to the study are included in the article or uploaded as supplementary information. All data relevant to the study are included in the article.

**ORCID iDs**
George Peat http://orcid.org/0000-0002-0293-2456
Beth Fylan http://orcid.org/0000-0003-0599-4537
David K Raynor http://orcid.org/0000-0003-0306-5275
Liz Breen http://orcid.org/0000-0001-5204-1187
David Phillip Alldred http://orcid.org/0000-0002-2525-4854

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
