## [Reviewer comments · BMJ Open]

ARTICLE DETAILS

TITLE (PROVISIONAL)	Barriers and facilitators of successful deprescribing as described by older patients living with frailty, their informal carers, and clinicians. A qualitative interview study.
AUTHORS	Peat, George; Fylan, Beth; Marques, Iuri; Raynor, David; Breen, Liz; Olaniyan, Janice; Alldred, David

VERSION 1 – REVIEW

REVIEWER	Ailabouni, Nagham J. University of South Australia
REVIEW RETURNED	30-Sep-2021

GENERAL COMMENTS	This is a well worth topic to explore. The authors employed sound research methods to investigate the perspectives of participants. The article was an interesting read and I hope you revise the manuscript taking into account a couple of minor suggestions. 1) Pg3; lines 14-19: Please reword this paragraph to say that several tools to support deprescribing and process tools have been created. However to date, no implementation science frameworks exist specific to deprescribing. Such a framework may allow researchers to implement deprescribing, with the knowledge of contextual determinants in a specific setting, to Also please cite: Ailabouni NJ, Reeve E, Helfrich CD, Hilmer SN, Wagenaar BH. Leveraging implementation science to increase the translation of deprescribing evidence into practice. Res Social Adm Pharm. 2021 Jun 6:S1551-7411(21)00184-4. doi: 10.1016/j.sapharm.2021.05.018. Epub ahead of print. PMID: 34147372. 2) Pg 17; lines 37-48: I overall agree with the theme “emotional factors” related to deprescribing. However, I believe this section would be stronger if in addition to this, fear of consequences of what would happen after deprescribing medication is added to the explanation. This is a very commonly cited barrier in the literature. Even if this is not cited in this section as this refers to the study’s results, it would be good to mention and discuss this point in the discussion section of the article.
--

REVIEWER	Ibrahim, Kinda Southampton University, Academic Geriatric Medicine, Faculty of Medicine
REVIEW RETURNED	01-Nov-2021

GENERAL COMMENTS	The paper investigates really interesting topic related to the concept of deprescribing among older people living with frailty,
---

	their carers and clinicians in primary care. the paper is well written and describe clearly the methods used (framework analysis) and present the findings in an understandable way. However, I have few comments/suggestions to improve the paper 1- Abstract: the abstract line 22-23 says " Clinicians (n=14) were interviewed once (n=38 interviews)" the two figures are confusing so can authors please clarify!. The results in the abstract did no really explain what are the identified barriers and facilitators but instead explained how many themes and which domains of the TDF they fit under. to make the abstract more informative, I suggest to describe the main barriers and facilitators identified and move any description of the fit with TDF to the method section 2- Study design: was this qualitative study part of a larger trial? if so, then the authors should refer to the research programme and describe its aim ...etc 3- Data collection: data was collected by a range of researchers, can the authors explain how they think this might have an impact on the quality and content of the data collected? any procedures were taken to minimise the negative impact of having different researchers for data collection? When patients and their informal caregivers were interviewed did that happen in separate interviews or were they interviewed together? how this could have an impact on their accounts and sharing their experiences? 4- data analysis: did you use any software for data management and analysis? or did you use excel or Microsoft words for analysis? please clarify 5- PPI: can you give a brief description of the characteristics of your PPI member and their previous experience 6- Results: Table 3 can you check the participants IDs at the end of quotes as some are missing or inconsistent 7- Results: the theme emotional factors does not seem for me to capture the content well and i feel it is more driven from the TDF domains. I suggest to try to try to rename this theme to reflect the content and data better 8- Discussion: the authors mentioned that future interventions should target behaviour domains linked to environmental context and resources, social/professional role and identity, and skills (interpersonal). can they expand and explain more what they mean and what the interventions should include to do to achieve that! 8- A strength to the study was interviewing patients/carers who underwent a deprescribing process, however, the study participants were all involved in a deprescribing intervention/process so they may represent a biased sample who is more willing to deprescribing than older people in general who might have different barriers to report. this needs to be acknowledged as a limitation to the findings
--	---

VERSION 1 – AUTHOR RESPONSE

Reviewer: 2

1- Abstract: the abstract line 22-23 says " Clinicians (n=14) were interviewed once (n=38 interviews)" the two figures are confusing so can authors please clarify!.

We have amended the abstract to clarify the two figures. N=38 is representative of the total interviews undertaken including with patients, their informal carers, and clinicians.

The results in the abstract did not really explain what are the identified barriers and facilitators but instead explained how many themes and which domains of the TDF they fit under. To make the abstract more informative, I suggest to describe the main barriers and facilitators identified and move any description of the fit with TDF to the method section

We have amended the results section of the abstract to clearly outline examples of barriers/facilitators of deprescribing for patients living with frailty identified by the study.

2- Study design: was this qualitative study part of a larger trial? if so, then the authors should refer to the research programme and describe its aim ...etc

A statement has been added at the end of the 'study design' section, outlining the research programme within which the qualitative study was situated.

3- Data collection: data was collected by a range of researchers, can the authors explain how they think this might have an impact on the quality and content of the data collected? any procedures were taken to minimise the negative impact of having different researchers for data collection?

We believe that using more than one interviewer is a strength, however all interviewers used the same interview guide and regular discussions were held to compare experiences of the interviews and discuss data as it was collected. Sentence added to methods section.

When patients and their informal caregivers were interviewed did that happen in separate interviews or were they interviewed together? how this could have an impact on their accounts and sharing their experiences?

Patients and their informal carers were interviewed together, doing so supported these patients to participate in the study. However, we acknowledge this as a potential limitation, as interviewing carers and patients together could have limited collecting the negative experiences of both due to the other being present. We have stated this limitation in the discussion.

4- data analysis: did you use any software for data management and analysis? or did you use excel or Microsoft words for analysis? please clarify

Microsoft Excel was used to manage the data-this has been clarified in the methods section of the paper, under 'data analysis'.

5- PPI: can you give a brief description of the characteristics of your PPI member and their previous experience

The PPIE member was an experienced local lay contributor and a lay leader within the NIHR Yorkshire and Humber Patient Safety Translational Research Centre. We have stated this in the method section.

6- Results: Table 3 can you check the participants IDs at the end of quotes as some are missing or inconsistent

Table 3 amended to show consistent participant IDs.

7- Results: the theme emotional factors does not seem for me to capture the content well and i feel it is more driven from the TDF domains. I suggest to try to try to rename this theme to reflect the content and data better

The theme emotional factors has been renamed as 'habits and fears' to more accurately reflect the data. We have also added further content to the theme for clarity.

8- Discussion: the authors mentioned that future interventions should target behaviour domains linked to environmental context and resources, social/professional role and identity, and skills (interpersonal). can they expand and explain more what they mean and what the interventions should include to do to achieve that!

We have aimed to clarify this section of the discussion with the addition of the following:

'Of the 14 TDF domains, environmental context and resources, memory, attention, and decision processes and skills (interpersonal) were the most identified by this study. Consequently, future interventions may target these behaviour domains by considering implementing prompts and cues for both healthcare professionals and patients, address multi-disciplinary patient-centredness and the patient's role in the deprescribing process.

8- A strength to the study was interviewing patients/carers who underwent a deprescribing process, however, the study participants were all involved in a deprescribing intervention/process so they may represent a biased sample who is more willing to deprescribing than older people in general who might have different barriers to report. this needs to be acknowledged as a limitation to the findings

We have noted this study limitation in the discussion section of the paper.

Reviewer: 1

Competing interests of Reviewer: None to declare

Reviewer: 2

Competing interests of Reviewer: NA